# Electrostatic Complementarities of Glioblastoma-Resident T-Cell Receptors and Cancer Testis Antigens Linked to Poor Outcomes and High Levels of Sphingosine Kinase-2 Expression

**DOI:** 10.3390/biology12040575

**Published:** 2023-04-10

**Authors:** Miguel A. Arias, Konrad J. Cios, Dorottya B. Kacsoh, Bailey E. Montgomery, Joanna J. Song, Anishaa R. Patel, Andrea Chobrutskiy, Boris I. Chobrutskiy, George Blanck

**Affiliations:** 1Department of Molecular Medicine, Morsani College of Medicine, University of South Florida, Tampa, FL 33612, USA; 2College of Medicine, University of Central Florida, Orlando, FL 32827, USA; 3Department of Pediatrics, Oregon Health and Science University Hospital, Portland, OR 97239, USA; 4Internal Medicine, Oregon Health and Science University Hospital, Portland, OR 97239, USA; 5Department of Immunology, H. Lee Moffitt Cancer Center and Research Institute, Tampa, FL 33612, USA

**Keywords:** adaptive immune receptors, cancer testis antigens, chemical complementarity scoring, glioblastoma, sphingosine kinase-2

## Abstract

**Simple Summary:**

The chemical complementarity of glioblastoma, tumor-resident T-cell receptors and cancer testis antigens were associated with a worse outcome. Additionally, the high expression of immune marker and low expression of apoptosis genes were associated with a high T-cell receptor–cancer testis antigen chemical complementarity and a worse outcome. In sum, T-cell receptor recombination reads from exome files have the potential to aid in glioblastoma prognoses and may provide opportunities to detect unproductive immune responses.

**Abstract:**

Introduction. Glioblastoma (GBM) is the most aggressive primary brain tumor in adults. Despite a growing understanding of glioblastoma pathology, the prognosis remains poor. Methods. In this study, we used a previously extensively benchmarked algorithm to retrieve immune receptor (IR) recombination reads from GBM exome files available from the cancer genome atlas. The T-cell receptor complementarity determining region-3 (CDR3) amino acid sequences that represent the IR recombination reads were assessed and used for the generation of chemical complementarity scores (CSs) that represent potential binding interactions with cancer testis antigens (CTAs), which is an approach particularly suited to a big data setting. Results. The electrostatic CSs representing the TRA and TRB CDR3s and the CTAs, SPAG9, GAGE12E, and GAGE12F, indicated that an increased electrostatic CS was associated with worse disease-free survival (DFS). We also assessed the RNA expression of immune marker genes, which indicated that a high-level expression of SPHK2 and CIITA genes also correlated with high CSs and worse DFS. Furthermore, apoptosis-related gene expression was revealed to be lower when the TCR CDR3-CTA electrostatic CSs were high. Conclusion. Adaptive IR recombination reads from exome files have the potential to aid in GBM prognoses and may provide opportunities to detect unproductive immune responses.

## 1. Introduction

Glioblastoma (GBM) remains incurable and has benefited little from new medical treatments since the current treatment approach was established in 2002 [1], although there have been some surgical innovations since that time. Because of the very limited success of current approaches, more recently emphasis has been placed on establishing GBM subtypes with the expectation of moving more in the direction of personalized medical treatments. In particular, the four subtypes of proneuronal, neural, classical, and mesenchymal [2] GBM have gained widespread acceptance with regard to consistent mutation profiles. Meanwhile, immune checkpoint blockade (ICB) has been a disappointment since the first attempts in 2013, with several exceptions related to subdividing GBM categories and with an eye towards one exemplary case of a mutator phenotype that results in possibly complete but, at least, very long-term remissions [3].

Several reasons have been proposed for the lack of improvement using ICB. Glioblastomas contain relatively few T-cells and an abundance of tumor-associated macrophages (TAMs), which enhance tumor development and promote treatment resistance. In contrast, other aggressive tumors have high T-cell infiltration and tend to be more responsive to ICB. Another explanation for GBM’s lack of responsiveness to ICB could be due to chemotherapy and corticosteroids, which are commonly used to manage GBM symptoms. For example, the immunosuppression caused by chemotherapy could negatively impact the effectiveness of ICB. Additionally, corticosteroids are commonly prescribed to manage cerebral edema and are anti-inflammatory, which would likely decrease lymphocyte tumor infiltration. GBM tumors can also adapt to ICB by upregulating different checkpoint protein ligands [4]. 

The above considerations indicate a need for additional immunological parameters for personalized treatments, including ICB treatments and possibly other immunotherapies or even anti-inflammatory therapies [5]. Specifically, for GBM, cancer testis antigens (CTAs) have come to light as possible targets for immunotherapy [6], with one very relevant indication that such CTAs can be specifically detected in GBM patients and not in healthy controls [7]. CTAs are generally thought to be commonly expressed in many cancers due to increased levels of gene demethylation in cancer cells, either indirectly, due to the inability to reassemble heterochromatic regions representing the CTA genes because of rapid cell division, or due to other, as yet not fully appreciated stem cell-like features of cancer cells that specifically reduce heterochromatin in regions of CTA genes. Thus, in this report, we have applied a recently developed algorithm for assessing the chemical complementarity between immune receptors and CTAs [8] to assess the relationship between such complementarity and DFS and GBM-related gene expression, with results revealing a novel connection between sphingosine kinase-2 (SPHK2) expression and an apparently failed immune response. 

## 2. Methods

### 2.1. Recovery of the TCGA-GBM (phs000178) IR Recombination Reads

The recovery of the TRA and TRB recombination reads from the GBM exome (WXS) files was per NIH dbGaP approved protocol number 6300, and the WXS file mining for these recombination reads has been extensively described [9,10,11]. The software for recombination read recovery, including readme files, is available at https://github.com/bchobrut-USF/blanck_group and a container version is available at https://hub.docker.com/r/bchobrut/vdj. An updated version of the software, with some added technical conveniences, is available at https://github.com/kcios/2021. The data representing the entire collection of TRA and TRB recombination reads extracted from the TCGA-GBM WXS files are available in supporting online material (SOM) Appendix A. The nucleotide sequences themselves are not available due to controlled-access restrictions; however, these reads will be made available by the corresponding author to investigators with proper dbGaP controlled-access approvals. 

### 2.2. Construction and Use of the Adaptive Match Web Tool

The chemical complementarity scoring is based on [8] and was facilitated by the construction of an original web tool, adaptivematch.com [5,12,13], which is publicly accessible and has been extensively benchmarked in [13]. The electrostatic complementarity scores (CSs) are generated by aligning the adaptive IR CDR3 AAs with a candidate antigen peptide sequence. The scoring process is described extensively in [8], including in the SOM of ref. [8], which includes algorithmic details and an mp4 video representing a user-friendly description. Basically, a positive charge in one entity directly across from a negative charge in the other entity would increase the score significantly, whereas proximity without direct alignment would represent a reduced contribution to a complementarity score increase. The adaptivematch.com web tool includes instructions for preparation of input files. Additionally, three example input files and two example output files are provided in the SOM as Appendix A, representing CS calculations with the CDR3s traceable to the TCGA-GBM WXS files. 

### 2.3. Gene Expression Analysis 

The TCGA-GBM gene expression assessments were performed using RNAseq values available via cbioportal.org. The RNAseq values were plotted against the electrostatic CSs to determine Pearson’s correlation coefficients and *p*-values. Again, this process is facilitated by adaptivematch.com, with the uploading of the csv files as detailed in the web tool instructions for preparation of input files containing RNAseq values and case IDs. For this study, the RNAseq values represented the Firehose legacy version of the TCGA-GBM dataset.

### 2.4. Survival Analyses

The Kaplan–Meier analyses were verified using the cbioportal.org web tool and GraphPad Prism as described [14,15,16].

### 2.5. Analysis of the Clinical Proteomic Tumor Analysis Consortium (CPTAC, phs001287) Dataset

To assess the correlation of CIITA RNAseq values with the CSs for TCR CDR3-SPAG9 Fragment 6, first, the adaptive IR recombination reads were obtained from the GBM subset of the CPTAC-3 study, by downloading the RNAseq files representing that dataset to USF research computing, via dbGaP approved protocol number 31752, and processing the RNAseq files with the software described and provided above, via the github links. Then, the CIITA RNAseq values were obtained by merging the files for each case ID for open access, “STAR counts”, available via the genomic data commons web tool. 

## 3. Results

To determine whether the electrostatic CSs (Methods) for GBM tumor-resident TCR CDR3s and CTAs could represent survival distinctions, we tested a set of CTAs, with results indicating that the upper 50th percentile of the TCR CDR3-CTA CSs for SPAG9, GAGE12F, and GAGE12G represented a worse DFS probability (Figure 1; Appendix A). To further investigate these results, we assessed the RNA expression of a panel of immune marker genes for the correlation of their expression levels with the electrostatic CSs, with results indicating that CIITA and SPHK2 expression correlated with the TCR CDR3-CTA CSs (Figure 2), while expression of the B-cell marker gene CD19 inversely correlated with the CSs (Table 1). 

We next considered the possibility that GBM tumor samples representing higher TCR CDR3-CTA CSs and worse DFS would also have a lower level of expression of apoptosis-related genes. We thus tested a panel of 22 apoptosis-effector genes and, most strikingly, the results indicated that AIFM3, which has a highly brain-specific expression pattern (http://genome.ucsc.edu/cgi-bin/hgGene?hgg_gene=ENST00000399167.6&hgg_chrom=chr22&hgg_start=20965107&hgg_end=20981360&hgg_type=knownGene&db=hg38), inversely correlated with the high TCR CDR3-CTA CSs (Figure 3, Table 2). 

We next divided the SPAG9 AA sequences into approximately 18 equal segments and tested each segment for a correlation of the TCR CDR3-SPAG9 fragment CSs with survival, with the results indicating that the SPAG9 Fragment 6 (KHIEVQVAQETRNVSTGSAENEEKSEVQAIIESTPELDMDKDLSGYKGSSTPTKGIENKA) in particular reproduced the strong inverse association of the TCR CDR3-CTA CSs with DFS and reproduced the same immune marker and apoptosis marker correlations, or inverse correlations (Figure 4; Table 3 and Table 4).

We next considered the possibility that the immune gene expression markers could represent survival distinctions for the entire GBM dataset rather than only for the samples with TRA and TRB recombination read recoveries. We focused on CIITA, in the immune marker case, because CIITA represented the most statistically significant correlations with the TCR CDR3-CTA CSs. The results indicated that upper and lower 50th percentiles for CIITA RNAseq values indicated a DFS distinction, with worse DFS probability associated with high CIITA RNAseq values (Figure 5).

Finally, we sought to replicate the correlation of the TCR CDR3-SPAG9 Fragment 6 CS with the CIITA gene expression using the CPTAC GBM dataset. In this case, the CSs were based on TCR recombination reads obtained from GBM RNAseq files, rather than WXS files, which were the source of the TCR recombination reads for the TCGA-GBM dataset analyses above. The results indicated that the CPTAC GBM TCR CDR3-SPAG9 Fragment 6 CSs did indeed correlate with the expression of CIITA (R value = 0.388, *p*-value = 0.0002). As indicated in Methods, this assessment was facilitated by the use of the adaptivematch.com web tool. The input files for the web tool for this CPTAC-related analysis are in the SOM as Appendix A. The output data are available in Appendix A. 

## 4. Discussion

The data above indicated that a higher electrostatic CS for TCR-CDR3-GBM CTAs and higher CIITA expression negatively correlated with GBM DFS probabilities, which emphasizes the need to more fully understand the potential negative impacts of the immune system on outcomes. While it is clear that an anti-tumor immune response in certain settings will facilitate tumor eradication, it is equally clear that certain inflammatory settings predispose to cancer growth. AIFM3, which is expressed almost exclusively in the brain, was expressed at a higher level in tumors where the TCR CDR3-CTA CSs were low, which is consistent with the idea that a high TCR CDR3-CTA CS is not representative of tumor cell killing by T-cells, i.e., lower DFS probabilities were consistent with the highly likely lack of apoptosis in the tumor samples. Thus, the important question becomes, is the inflammatory setting indicated by high TCR CDR3-CTA CSs supporting rather than reducing tumor growth?

The above results have two important limitations. First, there is no other reported approach, other than the in silico approach reported here, that supports the in vivo interaction of a TCR CDR3 with the CTAs that were identified with this in silico immunoinformatics approach. Second, there was no identification of the source of CIITA, which was identified as a pan-GBM dataset biomarker for a worse DFS probability. However, while additional work is needed to support the in silico indications of TCR and CTA binding, the algorithm used here will likely have use for patient risk stratification. As for CIITA, generally, when CIITA is expressed by tumor cells, which is very common in melanoma, it has a strongly anti-apoptotic function [17], consistent with the worse outcome reported here. However, CIITA, of course, could also be expressed by microenvironment antigen-presenting cells, consistent with a strong TCR CDR3–CTA interaction.

The data above also indicated that a higher level of SPHK2 was associated with the higher TCR CDR3-CTA CSs. Sphingosine 1-phosphate (S1P) plays a role in various cellular processes, including cell survival, proliferation, and apoptosis. Both mammalian sphingosine kinases, SPHK2 and sphingosine kinase 1 (SPHK1), generate S1P from a sphingosine precursor [18]. Both SPHK2 and SPHK1 have the capacity to facilitate tumorigenesis. However, SPHK2 has a more complex role, which is dependent on its subcellular localization. When SPHK2 is translocated to the plasma membrane, pro-proliferative signaling occurs, while translocation to internal organelles enhances anti-proliferative functions [18]. The cellular localization of SPHK2 to internal organelles is mediated by cytoplasmic dynein 1. It has been shown that GBM demonstrates lower cellular cytoplasmic dynein 1, and in turn, higher SPHK2 in the plasma membrane. In addition, it should be kept in mind that sphingosine-1-phosphate is a chemoattractant for T-cells [19,20], raising the question of whether SPHK2-generated sphingosine-1-phosphate in turn leads to a high infiltration of T-cells and ultimately to a pro-tumor, inflammatory environment?

## 5. Conclusions

A significant portion of GBM patients is likely to have only non-functional adaptive immune responses that may be actively supporting tumor growth.

## Figures and Tables

**Figure 1 biology-12-00575-f001:**
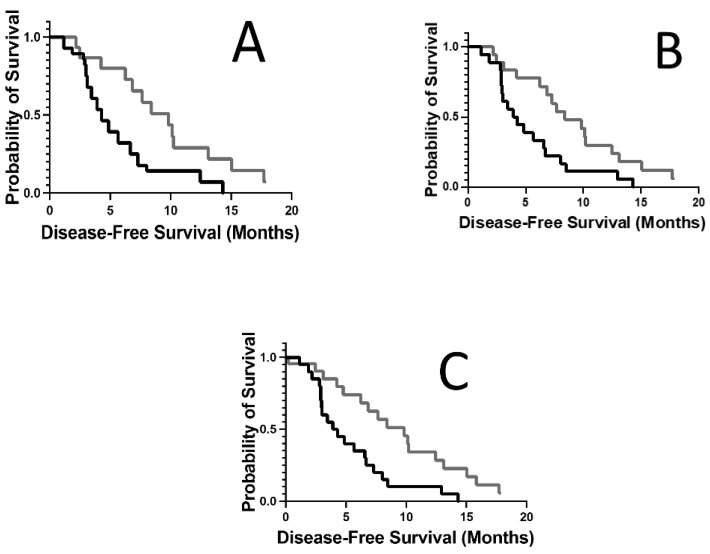
Kaplan–Meier (KM) analyses comparing disease-free survival (DFS) probabilities for case IDs representing the upper and lower 50th percentile TCR CDR3-CTA complementarity scores (CSs). (**A**) KM analysis of case IDs representing the upper (black line, n = 17) and lower (grey line, n = 18) 50th percentile TRA or TRB CDR3-SPAG9 CS groups. (**B**) KM analysis of case IDs representing the upper (black line, n = 19) and lower (grey line, n = 21) 50th percentile TRA or TRB CDR3-GAGE12F CS groups. (**C**) KM analysis of case IDs representing the upper (black line, n = 22) and lower (grey line, n = 22) 50th percentile TRA or TRB CDR3-GAGE12G CS groups.

**Figure 2 biology-12-00575-f002:**
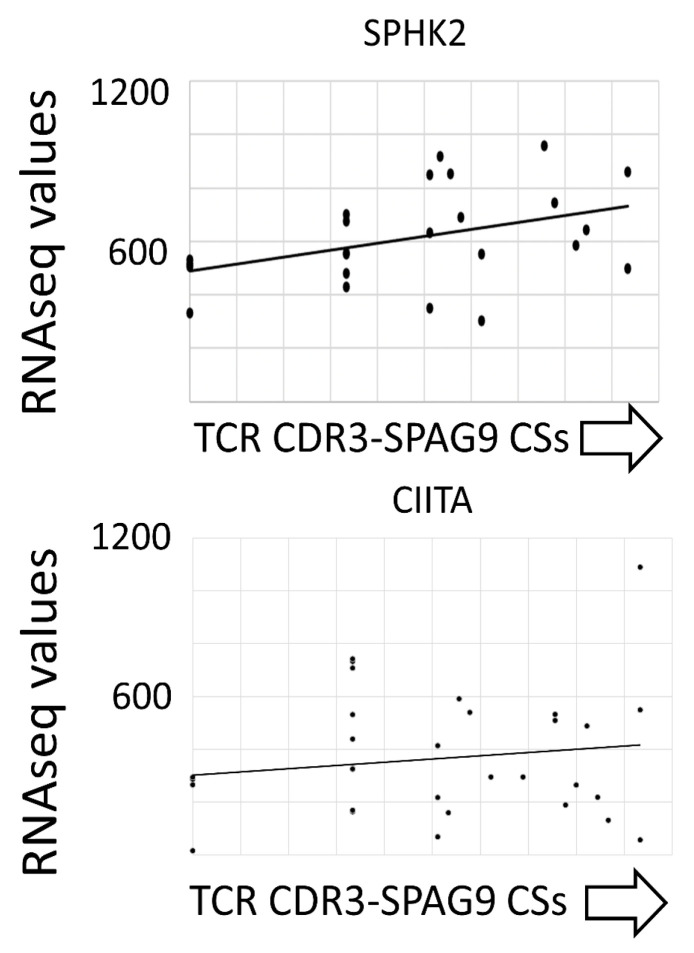
Correlation of SPHK2 and CIITA gene expression with TCR CDR3-SPAG9 electrostatic CSs. (See also Table 1; Appendix A).

**Figure 3 biology-12-00575-f003:**
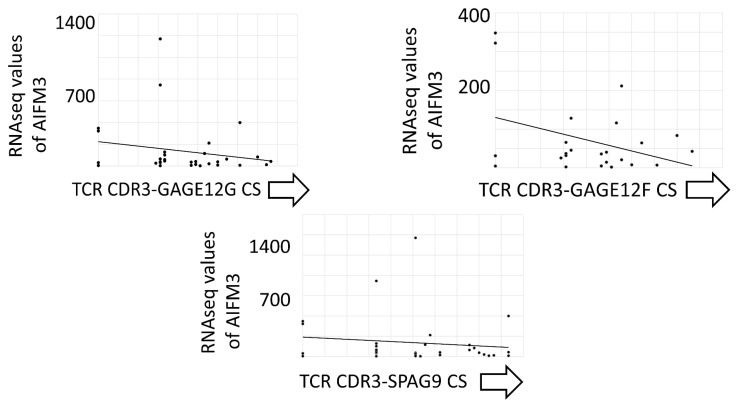
Inverse correlation of AIFM3 gene expression and electrostatic CSs for TCR CDR3-CTAs, SPAG9, GAGE12G, and GAGE12F. (See also Table 2; Appendix A).

**Figure 4 biology-12-00575-f004:**
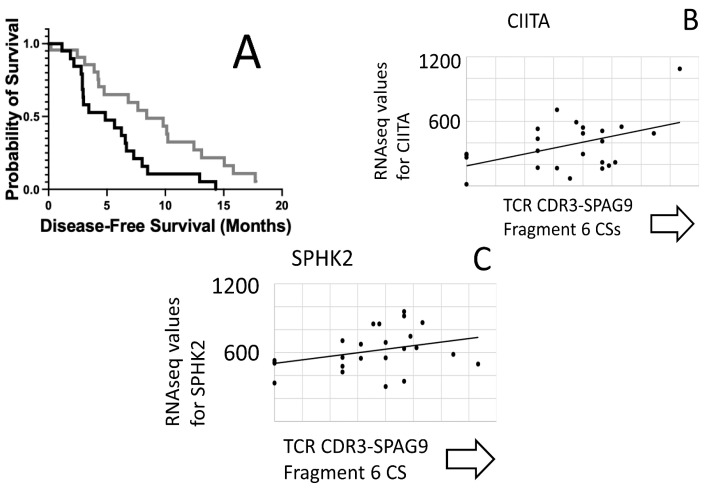
KM analysis comparing DFS for case IDs representing the upper and lower 50th percentile for the TCR CDR3-SPAG9 Fragment 6 electrostatic CSs. (**A**) The case IDs represent the upper 50th percentile electrostatic CS group (black line, n = 21) and lower 50th percentile electrostatic CS group (grey line, n = 23). (**B**,**C**) Scatter plots showing a correlation of the expression of SPHK2 and CIITA genes and electrostatic CSs for the TCR CDR3-SPAG9 Fragment 6. See also Table 3; Appendix A.

**Figure 5 biology-12-00575-f005:**
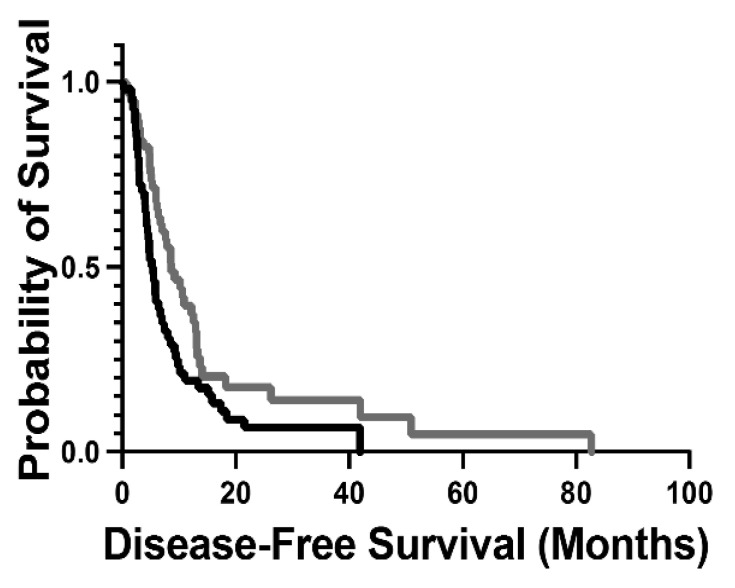
KM analysis comparing DFS for case IDs representing the upper (black line) and lower 50th percentile (grey line) CIITA RNAseq values for the entire TCGA-GBM dataset. (logrank *p*-value = 0.013; Appendix A).

**Table 1 biology-12-00575-t001:** Immune marker gene expression correlations with GBM TCR CDR3-CTA electrostatic CSs for SPAG9, GAGE12F, and GAGE12G.

(See also Figure 2)			
CTA	Immune Marker Gene	Pearson’s Coefficient	*p*-Value
GAGE12F	CIITA	0.486	0.0160
GAGE12G	CIITA	0.486	0.0160
SPAG9	CD19	−0.432	0.0348
SPAG9	CIITA	0.427	0.0373
SPAG9	SPHK2	0.408	0.0477
GAGE12G	SPHK2	0.406	0.0492
GAGE12F	SPHK2	0.406	0.0492
GAGE12F	CD19	−0.377	0.0691
GAGE12G	CD19	−0.377	0.0691

**Table 2 biology-12-00575-t002:** Apoptosis-effector gene expression, inverse correlations with GBM TCR CDR3 CTA CSs for SPAG9, GAGE12F, and GAGE12G.

CTA	Apoptosis Gene	Pearson’s Coefficient	*p*-Value
GAGE12F	COX7A2L	−0.438	0.0324
GAGE12G	COX7A2L	−0.438	0.0324
SPAG9	COX7A2L	−0.426	0.0378
SPAG9	AIFM3	−0.425	0.0385
SPAG9	UQCRC2	−0.408	0.0480
GAGE12G	UQCRC2	−0.395	0.0561
GAGE12F	UQCRC2	−0.395	0.0561
GAGE12G	AIFM3	−0.370	0.0747
GAGE12F	AIFM3	−0.370	0.0747

**Table 3 biology-12-00575-t003:** Immune marker gene expression correlations with GBM TCR CDR3-SPAG9 Fragment 6 electrostatic CSs.

Immune Marker Gene	Pearson’s Coefficient	*p*-Value
CIITA	0.465	0.0222
CD19	−0.396	0.0553
SPHK2	0.340	0.104

**Table 4 biology-12-00575-t004:** Apoptosis effector gene expression, inverse correlations with GBM TCR CDR3-SPAG9 Fragment 6 electrostatic CSs.

Apoptosis Effector Gene	Pearson’s Coefficient	*p*-Value
COX7A2L	−0.473	0.0195
AIFM3	−0.451	0.0268
UQCRC2	−0.429	0.0363

## Data Availability

The data presented in this study are available in the supporting online materials or are publicly available at cbioportal.org.

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
