# Peer review of "Electrostatic Complementarities of Glioblastoma-Resident T-Cell Receptors and Cancer Testis Antigens Linked to Poor Outcomes and High Levels of Sphingosine Kinase-2 Expression"

_biology, 2023, doi:10.3390/biology12040575_

Round 1
Reviewer 1 Report
Arias et al presented an in silico analysis on the available database of NIH dbGap and extracted glioblastoma exome files and showed specific fragment 6 of SPAG9 (CTA antigen) can be matched by specific T cell receptor repertoire in the sequenced samples and can predict the outcome and expression of apoptotic markers. Although this evaluation is unique but still rather preliminary and not confirmed experimentally at any level which makes it difficult for clinical practicality or translational part. There are lots of ambiguous methodologies which need clarification and justification, including the list of selected immune marker genes used and also the 22 apoptotic markers. No full data presentation about all of these complete lists has been provided. It seems the data of TCR is the extraction of 6 samples (10A, B, C and 01 A, B, 02A)! how representative this database can be regarding the generalization of this observation, if the filename is representative of each sample the bias of including several TCRs (some with 51 recorded unique TCR) from one file versus only 1 TCR from other files is not clear. How many other glioblastoma files were downloaded without any rearranged TCR and is that because of the sequencing method or the depth of the reads? Still, the final hypothesis is not clear how the CTA antigen associated with specific TCR can be linked with apoptotic markers like AIFM3 a schematic figure may help the authors to conclude their findings. And the most important step at least this hypothesis should be tested in real glioblastoma patients T cells confronting with GB cell lines and evaluating the apoptotic markers suggested with or without overexpression of SPAG9.
Author Response
Reviewer 1
First, authors thank the reviewer for the time spent and are well aware of the difficulty in finding suitable reviewers.
Arias et al presented an in silico analysis on the available database of NIH dbGap and extracted glioblastoma exome files and showed specific fragment 6 of SPAG9 (CTA antigen) can be matched by specific T cell receptor repertoire in the sequenced samples and can predict the outcome and expression of apoptotic markers. Although this evaluation is unique but still rather preliminary and not confirmed experimentally at any level which makes it difficult for clinical practicality or translational part.
Authors’ response: Given limitations and expertise, the assessment described in the manuscript is the current limit of our contribution. Our submission was meant to encourage others, especially others with expertise outside the area of bioinformatics and computational modeling of data, to consider the results for the types of approaches indicated by the reviewer. We also would like to add that, in silico assessments may provide compelling considerations not detectable by a specific in vitro or in vivo approach. It is possible a simple test of the binding of a CDR3 to antigen may not provide an impressive result but that such binding occurring in a patient may be beneficial for reasons that have yet to be discovered. Finally, the adaptivematch.com algorithm and output was substantially benchmarked in PMID 35987136. Also, authors note that a disease-free assessment represents a higher standard, and a more likely parameter to be associated with tumor immune activity, than does overall survival.
There are lots of ambiguous methodologies which need clarification and justification, including the list of selected immune marker genes used and also the 22 apoptotic markers. No full data presentation about all of these complete lists has been provided.
Authors response: The gene lists simply represent a relatively comprehensive collection of genes representing the cancer setting and have been consistently applied in past projects, beginning with PMID 24291030. No list would be perfect or completely comprehensive. With regard to full data presentation, we have now made it clear that no significant results were obtained for other genes besides what is indicated in the manuscript. However, it cannot be emphasized enough that AIFM3 is brain specific and indeed this is the gene that is active with longer disease-free periods.
It seems the data of TCR is the extraction of 6 samples (10A, B, C and 01 A, B, 02A)! how representative this database can be regarding the generalization of this observation, if the filename is representative of each sample the bias of including several TCRs (some with 51 recorded unique TCR) from one file versus only 1 TCR from other files is not clear.
Authors’ response: The reviewer is mistaken with regard to the numbers, including the reference to 10A, B, C, which represent blood and which were not used. Also, 02A was not used. Accurate numbers representing above categories can be obtained by examining Table S1, the total adaptive IR recovery list; and Table S3, the adaptivematch.com input file, representing the TCR (TRA + TRB) CDR3s for the complementarity score calculations. First, by filtering Table S1 on 01A and 01B, and TRA and TRB, there are 149 tumor resident CDR3s available from this TCGA-GBM dataset. All of those were placed into the adaptive match input file, Table S3. By removing duplicate case IDs from Table S3, it can be seen that 59 cases are represented. The range of CDR3 quantities for these 59 cases can be determined by using the Excel COUNTIF function. That range is 1 -15 CDR3s per case. These numbers are small in a certain sense. However, per authors PMID 34298587, small numbers of CDR3 can be reflective of consistency over distinct databases of CDR3s taken from different genomics datasets. This is likely because, as an immune repertoire, PCR based approach can show, the vast majority of amplified recombinations represent in most samples a very small number of CDR3s. Thus, by taking CDR3s from genomics files, one is likely taking the most common CDR3s, and indeed this would explain the correspondence of results seen in separate genomics file databases in PMID 34298587. Furthermore, GBM in particular is not highly immunogenic, keeping the numbers lower than might be expected for other cancers types. And it has to be noted that DFS information was not available for all 59 cases with TRA or TRB CDR3s. The actual number of cases used for the KM analyses are in those figures. The bottom line is that, the algorithm presented in our manuscript is novel, is capable of distinguishing patient groups, and is likely, but admittedly not certainly, reflective of patient immune responses to GBM.
How many other glioblastoma files were downloaded without any rearranged TCR and is that because of the sequencing method or the depth of the reads?
Authors’ response: The TCGA-GBM dataset has 433 cases with disease free survival data, as can be noted by the list in adaptivematch.com input Table S4.
Still, the final hypothesis is not clear how the CTA antigen associated with specific TCR can be linked with apoptotic markers like AIFM3 a schematic figure may help the authors to conclude their findings.
Authors’ response: There is no formal mechanistic linkage. This is a patient-based, correlative study. There is simply the consistency of a brain-specific apoptosis inducing factor being expressed at a significantly higher level among the cases where there is better disease-free survival. This makes sense, because with higher apoptosis, less disease detection would be expected.
And the most important step at least this hypothesis should be tested in real glioblastoma patients T cells confronting with GB cell lines and evaluating the apoptotic markers suggested with or without overexpression of SPAG9.
Authors’ response: Our group lacks that expertise, but again, it is important to note that one negative result in vitro would not automatically indicate that an in vivo scenario is not occurring. Closing in on a complete picture of in vitro, in vivo, and patient-centered processes is a large job for multiple specialties. Having said that, authors’ stand by the inevitable biomarker value of the work as presented, in addition to the “novel algorithm” based consistency with, but not certainty of immune mechanisms.
Reviewer 2 Report
This a timely and important study for those of us working in the field of glioma immunology. The work was carried out very well, the data presented well, and conclusions are sound. There are a few minor grammatical and/or miss-typed symbols which will no doubt be picked up by the proofing process post-acceptance and type setting.
Author Response
Reviewer 2
This a timely and important study for those of us working in the field of glioma immunology. The work was carried out very well, the data presented well, and conclusions are sound. There are a few minor grammatical and/or miss-typed symbols which will no doubt be picked up by the proofing process post-acceptance and type setting.
Author response: First, we appreciate the time spent by the reviewers, and authors are well aware finding reviewers is not easy. Given the comments for this reviewer, not corrections or changes are required, other than what would occur in review of the proof. We have however done another careful read through before returning the revision and corrected any minor errors that we have noticed.